# Juvenile Primary Sjögren Syndrome in a 15-Year-Old Boy with Renal Involvement: A Case Report and Review of the Literature

**DOI:** 10.3390/diagnostics14030258

**Published:** 2024-01-25

**Authors:** Katerina Bouchalova, Hana Flögelova, Pavel Horak, Jakub Civrny, Petr Mlcak, Richard Pink, Jaroslav Michalek, Petra Camborova, Zuzana Mikulkova, Eva Kriegova

**Affiliations:** 1Department of Pediatrics, Faculty of Medicine and Dentistry, Palacky University and University Hospital, 779 00 Olomouc, Czech Republic; 2Department of Internal Medicine III-Nephrology, Rheumatology and Endocrinology, Faculty of Medicine and Dentistry, Palacky University and University Hospital, 779 00 Olomouc, Czech Republic; pavel.horak@fnol.cz; 3Department of Radiology, Faculty of Medicine and Dentistry, Palacky University and University Hospital, 779 00 Olomouc, Czech Republic; jakub.civrny@fnol.cz; 4Department of Ophthalmology, Faculty of Medicine and Dentistry, Palacky University and University Hospital, 779 00 Olomouc, Czech Republic; petr.mlcak@fnol.cz; 5Department of Oral and Maxillofacial Surgery, Faculty of Medicine and Dentistry, Palacky University and University Hospital, 779 00 Olomouc, Czech Republic; richard.pink@fnol.cz; 6Department of Clinical and Molecular Pathology, Faculty of Medicine and Dentistry, Palacky University and University Hospital, 779 00 Olomouc, Czech Republic; jaroslav.michalek@fnol.cz; 7Department of Pediatrics, Tomas Bata Regional Hospital, 762 75 Zlin, Czech Republic; petra.camborova@bnzlin.cz; 8Department of Immunology, Faculty of Medicine and Dentistry, Palacky University and University Hospital, 779 00 Olomouc, Czech Republic; zuzana.mikulkova@fnol.cz (Z.M.); eva.kriegova@fnol.cz (E.K.)

**Keywords:** juvenile primary Sjögren syndrome, renal involvement, male child patient, immunophenotyping, diagnostic criteria, therapy

## Abstract

Juvenile primary Sjögren syndrome (pSS) with renal involvement is extremely rare, reported approximately in 50 children, predominantly girls. Here, we present the first reported case of a male child with juvenile pSS with ocular surface disease (previously keratoconjunctivitis sicca), submandibular salivary gland involvement, and tubulointerstitial nephritis. First, two symptoms were clinically apparent at presentation. We illustrate here that kidney involvement in pSS should be actively looked for, as juvenile pSS may be associated with asymptomatic renal involvement. Immunophenotyping of peripheral blood cells using multicolor flow cytometry revealed at the time of diagnosis changes in both adaptive (T memory cells and B memory cells), and innate immunity (an increased activation of natural killer cells, as well as monocytes and neutrophils, and an increased representation of intermediate monocytes). Our case report points to the importance of kidney examination, early diagnosis and therapy in juvenile pSS, as well as highlights international collaboration to obtain more data for this rare disease.

## 1. Introduction

Swedish ophthalmologist Henrik Sjögren first published a description of Sjögren syndrome (SS) as a “keratoconjunctivitis sicca”. The incidence and prevalence in childhood remain unknown, and it is considered to be an underrecognised and underdiagnosed entity. Originally, SS was reported most often in menopausal women, and arthritis was a prominent feature of SS, accompanied by a raised erythrocyte sedimentation rate (ESR), anemia, and fever. Since this early SS description, the syndrome has been identified as a heterogeneous chronic multisystem autoimmune rheumatic entity. Juvenile primary SS (pSS) is characterized by inflammation of the exocrine glands. However, extraglandular or systemic features can also be part of the disorder. Compared to adults, eye and oral symptoms are rare in childhood. Clinical manifestations of SS might be heterogenous, including dry eyes (keratoconjunctivitis sicca, corneal ulcers, keratitis), dry mouth (leading to increased caries), major salivary gland swelling (parotid, submandibular), extraglandular manifestations (fatigue, arthritis, arthralgia/myalgia, Raynaud phenomenon), pulmonary (chronic cough, interstitial lung disease: nonspecific interstitial pneumonia, lymphocytic interstitial pneumonia, usual interstitial pneumonia, small airways disease), renal (tubulointerstitial nephritis, renal tubular acidosis—with distal being more common) or neurological (peripheral neuropathy, central nervous system disease: demyelinating disease, neuromyelitis optica) involvement. SS is described as pSS when there is no association with other autoimmune diseases and as secondary SS (sSS) when there is another autoimmune disease present (i.e., systemic lupus erythematosus, mixed connective tissue disease, primary biliary cirrhosis, autoimmune thyroiditis). There is a female predominance both in adulthood and childhood, with a female-to-male ratio of 5:1 to 7:1 in juvenile pSS. In children, the mean age at initial symptoms is 10 years, with diagnosis at a mean of 12 years [1,2,3,4]. The diagnosis is based both on clinical and laboratory criteria. However, the American College of Rheumatology (ACR)-European League Against Rheumatism (EULAR) classification criteria are available only for adults [5]. Pediatric diagnostic criteria were proposed more than 20 years ago by Bartunkova et al. [6], based on a series of 8 children (7 females and 1 male), but they have not been validated on other cohorts yet. Recently, Devauchelle-Pensec et al. published a French national diagnostic and care protocol, including recommendations for SS in children [7]. Therapy is based on organ involvement, taking into account heterogeneity and different severity of individual patients. Pharmacotherapy encompasses predominantly local therapy, hydroxychloroquine, glucocorticoids, methotrexate, and biological disease-modifying antirheumatic drugs (bDMARDs) [7,8,9]. However, the lack of good-quality studies does not allow clinical recommendations for the treatment of SS with childhood onset [8]. Thus, prospective randomized clinical trials are needed. Here, we report a rare case of a male child with pSS with renal involvement, describing the clinical course and diagnostics, including immunophenotyping and therapy. Published cases of pSS with renal involvement are also reviewed.

## 2. Case Description

In 2019, a 15.5-year-old male presented with a 1-week history of fever up to 38.5 °C with cough and rhinitis during the last two weeks. Sterile pyuria lasting two weeks was also present. An otorhinolaryngologist at a regional hospital prescribed antibiotics for maxillary sinusitis and lymphadenopathy. Non-specific kidney abnormalities (multiple regions of parenchymal inflammation bilaterally) were found on computer tomography (CT), and he was referred to our center with suspicion of systemic disease.

Physical examination revealed redness of both eyes, narrowed eyelids, prominent neck enlargement, large and painful solid submandibular salivary glands and submandibular lymphadenopathy. Our patient suffered from ocular dryness (sensation of sand in the eyes). Schirmer test was used to detect ocular dryness. The ophthalmologist recommended local therapy (prednisolone, vitamin A, hyaluronic acid and ectoin) due to ocular surface disease. Ultrasound examination proved enlarged submandibular glands with a rough structure and heterogenous echogenicity (Figure 1A,B). A working diagnosis of “sicca syndrome” with renal involvement was established due to sterile pyuria, elevated protein/creatinine and albumin/creatinine ratios, slightly decreased glomerular filtration rate (GFR), and kidney imaging abnormality. Further, mild erythrocyturia was detected repeatedly. Microalbuminuria was also present. On the other hand, Astrup (acidobasis status) was repeatedly with normal pH. Laboratory findings revealed elevated ESR (up to 110 mm/1st hour), C-reactive protein (CRP, 35 mg/L), leukocytosis (13.88 × 10^9^/L) with neutrophilia, mild elevation in hepatic transaminases, hyperalbuminemia, polyclonal hypergammaglobulinemia, IgE elevation and hypovitaminosis D (25-hydroxyvitamin D level of 20.9 nmol/L). Antinuclear antibody (ANA), rheumatoid factor (RF), anti-extractable nuclear antigens (ENA), anti-Ro/SS-A and anti-La/SS-B autoantibodies were negative. Lupus anticoagulant was positive, while anti-cardiolipin IgG and anti-β2-glycoprotein 1 remained normal. Positron emission tomography (PET)-CT excluded malignancy. Further examination excluded IgG4-related, and other treatment/diseases (history of head and neck radiation treatment, active hepatitis C infection, acquired immunodeficiency syndrome, sarcoidosis, amyloidosis, graft versus host disease). Importantly, nephrology assessment revealed tubulointerstitial nephritis as the most probable diagnosis (sterile pyuria lasting four weeks, infection repeatedly excluded, while renal biopsy not carried out). Labial salivary gland biopsy revealed a focus score of 1, which met the 2016 ACR-EULAR SS classification criteria; however, a small amount of tissue did not allow the pathologist to exclude IgG4-related disease. Thus, a submandibular biopsy was performed and the findings supported the diagnosis of SS and excluded IgG4-related disease. Lymphocytic foci (i.e., foci with more than 50 lymphocytes) were found in the number of 5 per 4 mm^2^ of tissue in the submandibular salivary gland, resulting in a focus score of 3 (i.e., 2 or more lymphocytic foci per 4 mm^2^) (Figure 1C–I).

At the time of diagnosis, immunophenotyping of peripheral blood cells using multicolor flow cytometry revealed changes in T memory cells (decrease in central memory T cells, increase in PD1+ T cells) and B memory cells (decrease in CXCR3+ B cells) (Figure 2A). Regarding the innate immune cells, increased activation of natural killer (NK) cells (increased expression of the early activation marker CD69), as well as monocytes (increased number of CD64+ and TLR4+ monocytes) and neutrophils (increased expression of CD11b, CD54 and CD64) were detected, as well as an increased representation of intermediate monocytes (Figure 2B,C).

A diagnosis of primary pSS was established by ACR-EULAR classification criteria for adult patients [5], as well as with the proposed diagnostic criteria for juvenile pSS [6]. The patient met six criteria proposed by Bartunkova et al. (oral and ocular clinical symptoms and four laboratory parameters: elevated ESR, polyclonal hyperimmunoglobulinemia, histological proof of lymphocytic infiltration of salivary glands, and objective documentation of ocular surface disease/dryness), and two ACR-EULAR criteria (focal lymphocytic sialadenitis and Schirmer). Due to nephrology indication (remained sterile pyuria, elevated albuminuria, and slow decrease in GFR) corticoid therapy was started (three 250 mg pulses of methylprednisolone followed by prednisone). Within 3 days after initiation of corticoid treatment, GFR and albuminuria normalized and sterile pyuria disappeared. Glycosuria (5.6 mmol/L) with hyperglycemia appeared as side effects. Weight gain and hypertension had developed, and thus antihypertensives (amlodipine and losartan) were prescribed. Due to his age, he was discussed with an adult rheumatologist, and therapy was amended with hydroxychloroquine. After three weeks, the patient left the hospital in good status. The following care was altered due to the COVID-19 pandemic with successful telemedicine and reinforced collaboration with regional pediatricians. The patient’s status further improved and only borderline anemia persisted. He managed to lose the weight via correction of eating habits and overall lifestyle. Four years after diagnosis, the patient is in good status, without medication, undergoing the transition process to the adult rheumatology. He attends a school.

## 3. Discussion

Here, we report the first case of a male child with juvenile pSS with ocular surface disease, submandibular salivary gland involvement, and tubulointerstitial nephritis. Renal involvement of pSS is a rare disease manifestation in childhood with 50 cases described in the literature (Table 1) [6,10,11,12,13,14,15,16,17,18,19,20,21,22,23,24,25,26,27,28,29]. The largest published series by Basiaga et al. [29] encompassing 300 SS patients found 2 males among 20 patients with pSS and renal involvement (Table 1). Interestingly, renal involvement was found more frequently in children compared to adults (19.2% vs. 3.9%) (*p* = 0.005) in one study on SS including both pSS and sSS [25]. The same study described renal involvement to be present only in children without parotitis (*p* = 0.004) [25].

Kidney involvement should be actively looked for. Surprisingly, even one case with juvenile pSS detected by urine screening is described in the literature [13].

In adults, tubulointerstitial nephritis in SS is frequently accompanied by tubular syndromes—renal tubular acidosis, aminoaciduria, glycosuria with normal glycemia, Fanconi syndrome, and pseudo-Bartter syndrome. There were no above-mentioned syndromes detected in our patient.

Interestingly, no autoantibodies (ANA, ENA, anti-Ro/SSA, anti-La/SSB, RF) were detected in our pSS patient. In the pediatric pSS population, Virdee et al. reviewed a variability of positive serological markers, with anti-Ro/SSA, and anti-La/SSB antibodies between 36.4–84.6% and 27.3–65.4%, respectively. They even found a lower frequency of anti-Ro/anti-La in the male population. ANA and RF positivity varied in pediatric pSS populations between 63.6–96.2% and 27.3–75%, respectively [1].

Our patient suffered from ocular surface disease (ocular dryness). Ocular surface disease (previously keratoconjunctivitis sicca) and submandibular salivary gland enlargement are clinically apparent symptoms. Children diagnosed with pSS reported less dryness compared to adults [1]. In a study by Bartunkova et al., sicca syndrome was never seen during childhood, but it developed later in 3 out of 8 pSS patients [6].

In our patient, at the time of diagnosis, we also found changes in the immunophenotypes of circulating immune cells, particularly a decrease in central memory T cells, a decrease in CXCR3+ B cells, an increased proportion of intermediate monocytes, and increased activation of NK cells, monocytes, and neutrophils. To date, only a few studies have reported immune cell immunophenotypes in SS, mainly focusing on adults. Immunophenotyping of peripheral blood in adult pSS patients with low disease activity or in clinical remission revealed two subgroups of patients, characterized by distinct immune cell profiles [30]. Interestingly, patients with pSS and systemic lupus erythematosus had a similar immunologic architecture [30]. According to our knowledge, there is only a pilot study in 10 children with juvenile SS presented as a conference paper [31]. They detected elevated naïve and reduced frequencies of memory B cells and a dysregulation of the responder CD8+ T cell subpopulation compared to healthy controls [31]. The role of monocytes in SS has been supported by RNA-sequencing, where the transcriptomic profile of pSS patients has been shown to be enriched in intermediate and non-classical monocyte profiles [32,33]. Activated neutrophils have also been reported in SS patients [34,35], and it has been shown that they may contribute to inflammatory manifestations in pSS patients through induced extracellular trap formation (NETosis) [35]. Our data further support the key role of innate immune cells in the pathogenesis of pSS.

Therapy recommendations for adults have been developed by EULAR [9]. Recently, the French national diagnostic and care protocol includes therapeutic recommendations for SS in children [7]. Steroid and hydroxychloroquine therapies were found to be effective in treating renal disease [8].

SS could be associated with a higher risk of lymphoma [18]. The patient underwent PET-CT to exclude malignancy and is followed up with.

## 4. Conclusions

To the best of our knowledge, this is the first pediatric male case with tubulointerstitial nephritis, ocular surface disease (previously keratoconjunctivitis sicca), and submandibular salivary gland involvement. Our report highlights the importance of kidney examination, as well as early diagnosis and therapy. International collaboration is necessary to collect more data on juvenile pSS.

## Figures and Tables

**Figure 1 diagnostics-14-00258-f001:**
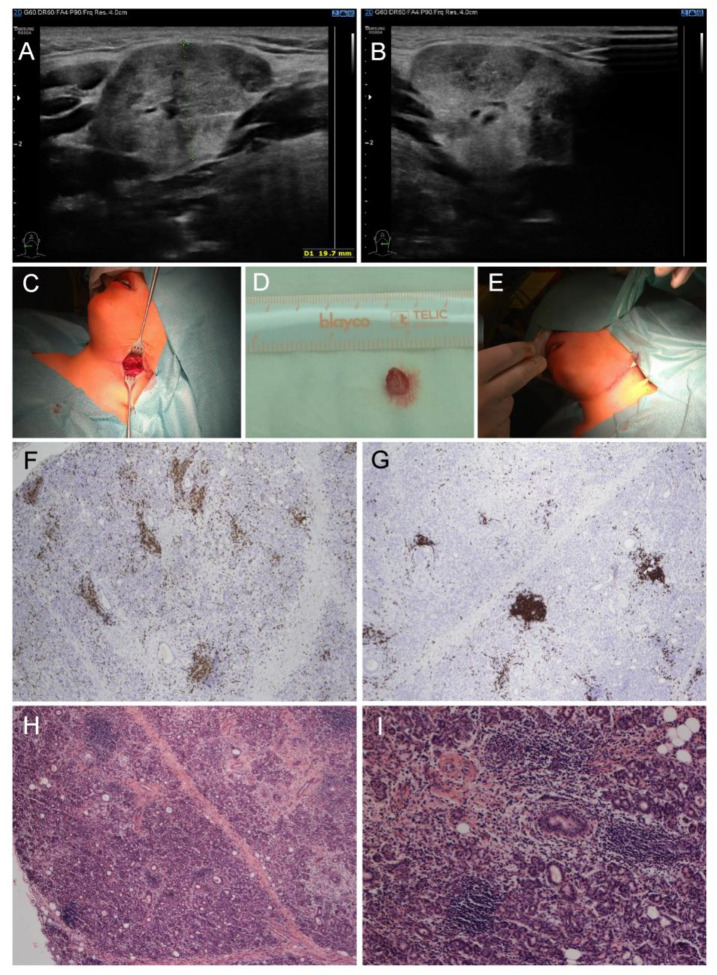
Clinical and pathological examinations of a child male with pSS at the diagnosis. (**A**,**B**) Ultrasound: enlarged submandibular glands with a rough structure and heterogenous echogenicity. (**C**–**E**) Macroscopic findings: (**C**) submandibular salivary gland before probatory excision, (**D**) bioptic material from hypertrophic submandibular gland, (**E**) direct suture in submandibular region. (**F**–**I**) Histopathological findings in submandibular salivary gland: multifocal lymphocytic sialadenitis: (**F**) Immunohistochemistry of CD3 antibody labeling T lymphocytes, (**G**) immunohistochemistry of CD20 antibody labeling B lymphocytes, (**H**) hematoxylin–eosin staining of multifocal lymphocytic sialadenitis (40×), (**I**) hematoxylin–eosin staining of multifocal lymphocytic sialadenitis (100×).

**Figure 2 diagnostics-14-00258-f002:**
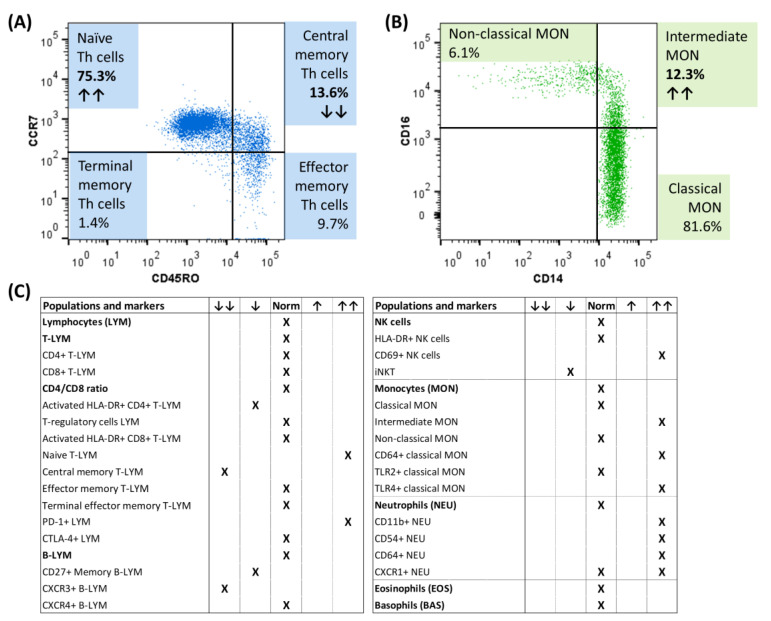
Flow cytometric immunophenotyping of circulating immune cells of a child male with pSS at the diagnosis. (**A**) Distribution of memory T cell subsets according to CCR7 and CD45RO expression. (**B**) Representation of monocyte subsets. (**C**) Overview of the representation of immune populations/subpopulations and their activation in our patient with pSS at the time of diagnosis. ↑↑ indicates an increased relative percentage of the presented subpopulation/increased activation, while ↓↓ indicates a decrease from the healthy controls; CTLA-4+ = cytotoxic T lymphocyte-associated antigen positive, CXCR3+ = chemokine receptor CXCR3 positive, CXCR4+ = chemokine receptor CXCR4 positive, PD-1+ = programmed cell death protein positive, iNKT = invariant NK T cells, TLR2+ = toll-like receptor 2 positive, TLR4+ = toll-like receptor 4 positive.

**Table 1 diagnostics-14-00258-t001:** Published cases with juvenile pSS with renal involvement.

References	Country	N of pSS pts with R Involvement	Age	Sex	Renal Involvement	Biopsy Yes/No
Shioji 1970 [10]	Japan	1 child	14	F	pRTA	yes, focal infiltration in interstitium
Chang 1995 [11]	Taiwan	3 with RTA	5	F	dRTA, HPP	no
			15	F	dRTA, HPP	no
			5	F	pRTA, HPP	no
Kobayashi 1996 [12]	Japan	1 with pSS	10	F	dRTA (nephritis)	yes
Yoshida 1996 [13]	Japan	1	13	M	HU, PU	yes, membranous GN
Tomiita 1997 [14]	Japan	3 with nephritis	NM	F *	nephritis	yes, IN
			NM	F *	nephritis	yes, IN
			NM	F *	nephritis	yes, IN
Zawadzki 1998 [15]	Poland	1 with RTA	15	F	dRTA	yes, IN, fibrosis, and hyalinization
Bartunkova 1999 [6]	Czech	3 with RTA	14	F	dRTA	NM
			16	F	RTA	NM
			10	F	RTA	yes, immunocomplex GN *
Ohlsson 2006 [16]	UK	1	8	F	dRTA	no
Pessler 2006 [17]	USA	2	10	F	pRTA	NM
			8	F	pRTA, dRTA	NM
Johnson 2007 [18]	UK	2	13	F	end-stage renal failure	yes, severe inflammation
			14	M	end-stage renal failure	yes, severe inflammation
Skalova 2008 [19]	Czech	1	16	F	dRTA	yes, chronic TIN
Maripuri 2009 [20]	USA	1 child	15	F	dRTA	TIN
Jung 2010 [21]	Korea	1	11	M	HU, PU, dRTA	yes, MPGN and IgA deposits
Kagan 2011 [22]	Russia	1	12	F	HU, PU, renal failure	yes, crescentic GN, TIN
Igarashi 2012 [23]	Japan	1	12	F	PU	yes, TIN
Bogdanovic 2013 [24]	Serbia	1, plus review	13	F	dRTA, nephrocalcinosis	yes, severe IN
Yokogawa 2016 [25]	USA	5 with R, 2/5 sSS	NM	F	nephrocalcinosis	NM
			NM	F	nephrocalcinosis	NM
			NM	F	RTA	NM
			NM	F	RTA	NM
			NM	F	nephritis	NM
Matsui 2016 [26]	Japan	1	9	F	RTA	NM
Zhao 2020 [27]	China	1	12	F	glycosuria	yes, tubulointerstitial damage
Kobayashi 2020 [28]	Japan	1	11	F	APSGN	yes, APSGN and IN
Basiaga 2021 [29]	8 countries **	2	NM	M *	PU *	no *
		10	NM	F *	PU *	NM
		5	NM	F *	IN *	NM
		3	NM	F *	RTA *	NM

APSGN = acute poststreptococcal glomerulonephritis; dRTA = distal RTA; F = female; GN = glomerulonephritis; HPP = hypokalemic periodic paralysis; HU = hematuria; IN = interstitial nephritis; M = male; MPGN = mesangial proliferative glomerulonephritis; NM = not mentioned; pRTA = proximal RTA; pSS = primary SS; pts = patients; PU = proteinuria; R = renal involvement; RTA = renal tubular acidosis; sSS = secondary SS; TIN = tubulointerstitial nephritis; * personal communication; ** USA, Brazil, Spain, Australia, Italy, Japan, Poland, Serbia.

## Data Availability

The datasets generated and analyzed during the current study are available from the corresponding author upon reasonable request.

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
