# Peer review of "Juvenile Primary Sjögren Syndrome in a 15-Year-Old Boy with Renal Involvement: A Case Report and Review of the Literature"

_diagnostics, 2024, doi:10.3390/diagnostics14030258_

Round 1
Reviewer 1 Report
Comments and Suggestions for Authors
A very interesting case presented with high quality and supported by amazing figures and literature. Sjögren syndrome is a rare autoimmune disease in childhood which is also possibly overlooked by clinicians. I would like to congratulate the authors for their motivation to diagnose the patient who even has no autoantibody positivity. I have only 2 minor concern:
1) The meaning of the first sentence of the abstract is not clear. Please, rephrase
2) I am wondering if the patient has undergone any test to detect any ocular dryness. Please, explain.
Reviewer 2 Report
Comments and Suggestions for Authors
The authors reported a juvenile case of Sjögren’s syndrome (SS) complicating tubulointerstitial nephritis, and stated the patient was compatible with 2016 ACR-EULAR SS criteria. However, there are several uncertain points about the diagnosis.
1. Existence of subjective sicca symptoms is required as inclusion criteria of 2016 ACR-EULAR SS criteria, which is not clear in Case Description.
2. Line 107 to 109: ‘Labial salivary gland biopsy was non-diagnostic, however, the subman- dibular one supported SS. Lymphocytic foci (i.e. foci with more than 50 lymphocytes) were found in the number of 5 per 4 mm2 of tissue.’ In 2016 ACR-EULAR SS criteria, ‘Labial salivary gland with focal lymphocytic sialadenitis and focus score of≧1 foci/4 mm2’ is stated.
3. How about results of ocular staining score?
4. How about anti-Ro/SS-A antibodies? Negative results of ANA do not always exclude possibility of positive anti-Ro/SS-A antibodies.
5. Did the authors all exclusion criteria in 2016 ACR-EULAR SS criteria? The explanation in Line 104 ‘Further workup excluded IgG4-related and other diseases.’ is not satisfactory.
6. Considering above, the authors had better consider the other classification criteria for SS, such as 2012 SICCA criteria and e 2002 American-European Consensus Group SS classification criteria
Round 2
Reviewer 2 Report
Comments and Suggestions for Authors
(No further comments)